# Analysis on the Return Period of "7.20" Rainstorm in the Xiaohua Section of the Yellow River in 2021

Shuangyan Jin *, Shaomeng Guo and Wenbo Huo

Hydrology Bureau of Yellow River Conservancy Commission, Zhengzhou 450004, China
* Correspondence: yrccjinzi@163.com

**Abstract:** The "7.20" rainstorm and flood disaster in 2021 occurred in Zhengzhou and adjacent areas of Henan province. According to the Maximum Rainfall Data of Different Periods and the "7.20" rainstorm data of the section from Xiaolangdi to Huayuankou of the Yellow River in 2021, i.e., thirteen kinds of automatic monitoring rainfall data in minutes and six kinds of manual monitoring rainfall data in hours, the rainfall frequency curves of two representative periods of ten reference stations are established using Pearson-III distribution. The return periods of "7.20" rainstorms with maximum 24 h greater than 300 mm and maximum 6 h greater than 100 mm are calculated. The results show that the influence of "7.20" rainstorms on the rainfall return period is obvious. For the ten reference stations, all the maximum 24 h rainfall of "7.20" rainstorms ranked in the first of the series. When establishing the frequency curve, if this value is considered, the largest return period occurs at Sishui station, with a return period of 486 years. Otherwise, the return period of Sishui, Mangling, Minggao, and Xicun stations will exceed 10,000 years. Among ten reference stations, the largest proportion of the maximum 6 h rainfall between "7.20" rainstorms and historical series is Minggao station. Taking Minggao station as an example, the return period is about 200 years when considering the "7.20" value to establish the frequency curve, otherwise it is about 3000 years. The results can provide technical support for the analysis of the rainstorm return period and the flood control operation in the lower Yellow River.

**Keywords:** return period; Pearson-III distribution; "7.20" rainstorm; Yiluo River basin; Xiaohua section





## 1. Introduction

Global warming and the acceleration of urbanization have made extreme weather events more serious and frequent in many countries and regions [1–3], and the frequency of extreme precipitation is still increasing [4]. From 17 to 23 July 2021, Henan province suffered a rare heavy rain in history, resulting in serious floods; especially, on 20 July, Zhengzhou suffered heavy casualties and property losses [5,6]. The "7.21" rainstorm in Beijing in 2012 caused urban waterlogging [7,8]. A rare heavy rain occurred in Shiyan and Xiangyang in Northwest Hubei from 4 to 6 August 2012 [9], and the maximum 24 h rainfall was 685.5 mm. An extreme rainstorm occurred in Jinan on 18 July 2007 [10]. The phenomenon of "seeing the sea in cities" has occurred repeatedly in major cities of China.

Scholars have fully studied extreme precipitation events in different regions from different timescales [11–13]. Manton et al. [14] found that where the number of precipitation days decreased, extreme heavy precipitation events increased. The multivariable hydrological frequency analysis method has been widely used in the risk analysis of urban flood disaster. For example, Zellou and Rahali [15] used the multivariable hydrological frequency analysis method to analyze the encounter risk of precipitation and tide level, the two main disaster-causing factors in the bouregreg estuary of Morocco, and the urban risk. The maximum entropy principle was applied to calculate the return period of extreme precipitation based on the Mann–Kendall test [16–18]. The climatic factors of rainstorm

and waterlogging in different regions are obviously different, but extreme weather leads to the increase of extreme precipitation events and local flooding frequency, which is a new challenge for different countries, cities, and regions.

The return period refers to an average of hydrological events that can occur in a number of years during long periods [19–21]. It is a design standard widely used in the planning program, operation and management of water conservancy, and hydropower projects and civil engineering [22–24]. It is mainly determined by the importance of engineering and the result of damage to the project of hydrological events [25–27]. Relevant scholars have conducted a lot of research on the return period of rainstorms using different methods, such as the Copula function [28–30], multi-mode coupling [31], risk analysis [32], annual maximum sampling [33], random rainstorm transplantation [34–37], the long-term comprehensive rainstorm formula based on the attenuation rainstorm characteristics [38–41], and so on. Ding et al. [28] used Copula functions to develop the multivariate joint distribution of annual maximum storms at Heyuan Station of Dongjiang River Basin, and the probability of different joint storm variables and the corresponding return periods were analyzed. The extreme precipitation frequency estimation based on the annual maximum series was revised by using the hydrometeorological regional L-moments method and Chow's equation for Jiangsu Province [33].

There have been a lot of studies on extreme precipitation and the rainstorm return period, and some experts have also studied the characteristics and causes of extreme rainstorm precipitation in Henan [42–45]. However, there is no clear understanding of the study on the return period of rainstorms in Xiaohua interval and Yiluo River Basin of the Yellow River. Especially when the current rainfall is the maximum value of the series, it is not clear whether the maximum value is considered when establishing the frequency curve.

For "7.20" rainstorms in the Xiaohua Section, the current rainfall of many stations is the maximum of the series. However, different results about the return period of this heavy rain were provided. Some people believe the return period is about 1000 years, while others believe it is more than 10,000 years.

When the current rainfall of a single station is the maximum of the series, should it be considered in establishing the Pearson-III frequency curve when analyzing rainfall recurrence? What is the difference between considering this value or not? Is this the main reason that causes the different conclusions of the recurrence?

To answer these questions, ten stations with long series rainfall observation data in the study area were selected as representatives. The Pearson-III distribution was adopted to fit the line and return periods of 42 rainfall stations with maximum 24 h rainfall more than 300 mm and 46 rainfall stations with maximum 6 h rainfall more than 100 mm were presented.

Therefore, the main objective of this work is to study the return period when the current rainfall is the maximum value of data series of analyzed rainfall stations. The results can provide technical support for the analysis of the rainstorm return period and flood control operation in the lower Yellow River.

## 2. Data and Method

### 2.1. Study Area

The Xiaohua Section of the Yellow River includes the main stream section from Xiaolangdi to Huayuankou, and the tributaries including Yiluo River and Qin River (shown in Figure 1).

The Yiluo River is an important primary tributary of the Yellow River, and one of the main sources of flood in the lower reaches of the Yellow River [46]. It originates from Luonan, Shaanxi Province, with a basin area of 18,881 $km^2$, involving 21 counties and cities in Henan and Shaanxi provinces. The controlling station of Yiluo River entering the Yellow River is Heishiguan hydrological station, above which the catchment area is 18,563 $km^2$. The main stream, Luohe River, is 446.9 km and the tributary Yihe River is 264.8 km.

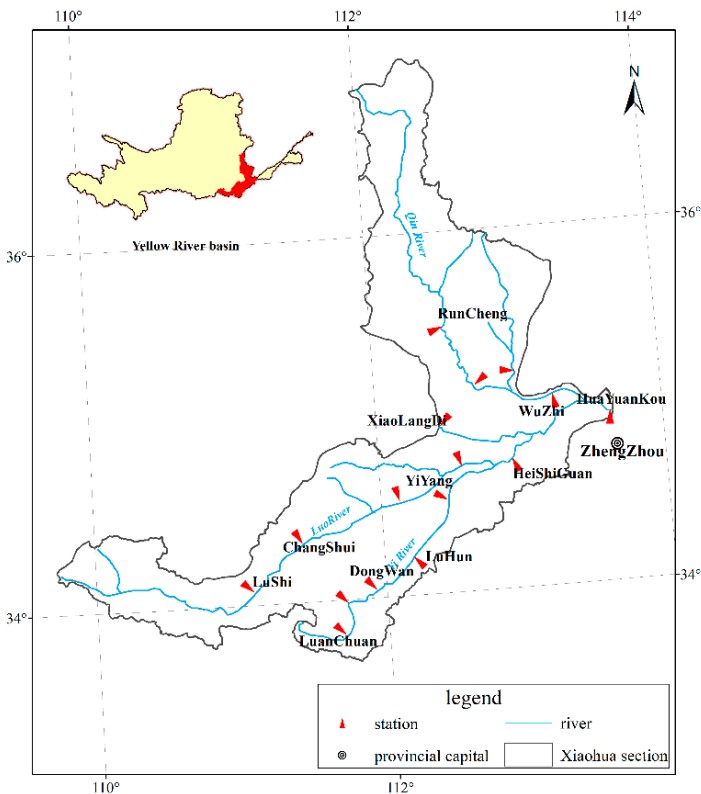

**Figure 1.** The location of the Xiaohua section of the Yellow River.

The Qin River is a primary tributary of the Yellow River located in the northern region of Xiaolangdi to Huayuankou section, with a basin area of 13,532 km². It originates from Qinyuan County, Shanxi Province. Wuzhi hydrological station is the controlling station of Qin River entering the Yellow River, above which the catchment area is 12,880 km² (shown in Figure 1).

### 2.2. Basic Data

According to Tables 1 and 2 of the Maximum Rainfall Data of Different Periods over the years, thirteen kinds of automatic monitoring rainfall data in minutes and six kinds of manual monitoring rainfall data in hours were used, and ten long-series rainfall stations were selected, including Xiaoguan, Sishui, Mangling, Jiulongjiao, Guangwu, Wulongmiao, Minggao, Lijiajie, Dongdi, and Xicun (shown in Figure 2). The annual maximum 24 and 6 h rainfalls were selected from the same tables over the years. The above data are shown in the Hydrology Yearbook of the Yellow River Basin [47].

**Table 1.** The top 20 daily rainfall events (mm).

| No. | Station | Date | Rainfall | No. | Station | Date | Rainfall |
|-----|---------|------|----------|-----|---------|------|----------|
| 1 | Fenggou | 20 July | 454.5 | 11 | Heluozhen | 20 July | 326 |
| 2 | Liuhe | 20 July | 429 | 12 | Xinzhong | 19 July | 322 |
| 3 | Duanhecun | 20 July | 424.5 | 13 | Shidonggou | 20 July | 316 |
| 4 | Huancuiyu | 20 July | 396.5 | 14 | Honghe | 19 July | 312.5 |
| 5 | Huancuiyu | 19 July | 390.5 | 15 | Zhulin | 20 July | 309 |
| 6 | Hegou | 20 July | 386 | 16 | Gongchuan | 19 July | 309 |
| 7 | Xiaoguan | 19 July | 366 | 17 | Gaoshan | 20 July | 308 |
| 8 | Xinzhong | 20 July | 362.5 | 18 | Zhulin | 19 July | 304.5 |
| 9 | Llijiaguan | 19 July | 360.4 | 19 | Shennan | 20 July | 298 |
| 10 | Xiayu | 19 July | 341.5 | 20 | Liangshuiquan | 19 July | 295 |

**Table 2.** The average rainfall in each region on July 18 to 22 in the Xiaohua section.

| Section | Covered Area under Different Rainfall | | | | | | | | | Area/km² | Mean Rainfall/mm |
|---|---|---|---|---|---|---|---|---|---|---|---|
| | 0~100 | 100~200 | 200~300 | 300~400 | 400~500 | 500~600 | 600~700 | 700~800 | 800~860 | | |
| Upper Baimasi | 7276 | 3964 | 629 | 21.8 | 0 | 0 | 0 | 0 | 0 | 11,891 | 102.6 |
| Upper Longmenzhen | 796 | 3264 | 1100 | 149 | 7 | 2 | 0 | 0 | 0 | 5318 | 165.3 |
| B, L~Heishiguan | 0 | 221 | 436 | 372 | 315 | 33.0 | 5.2 | 1.7 | 0 | 1384 | 315.7 |
| Upper Runcheng | 5272 | 1580 | 406 | 13.9 | 0 | 0 | 0 | 0 | 0 | 7273 | 88.4 |
| R~Wulongkou | 14 | 499 | 706 | 659 | 94.6 | 0 | 0 | 0 | 0 | 1972 | 266.4 |
| Upper Shanluping | 10 | 631 | 1619 | 439 | 350 | 0 | 0 | 0 | 0 | 3049 | 266.1 |
| W, S~Wuzhi | 0 | 0 | 48.1 | 412 | 119 | 7.83 | 0 | 0 | 0 | 586 | 364.7 |
| Xiaohua main stream | 0 | 157 | 778.3 | 1603 | 1065 | 391 | 205 | 119 | 23 | 4342 | 395.1 |
| Xiaohua Section | 13,368 | 10,317 | 5724 | 3669 | 1950 | 434 | 211 | 121 | 23 | 35,815 | 180.9 |

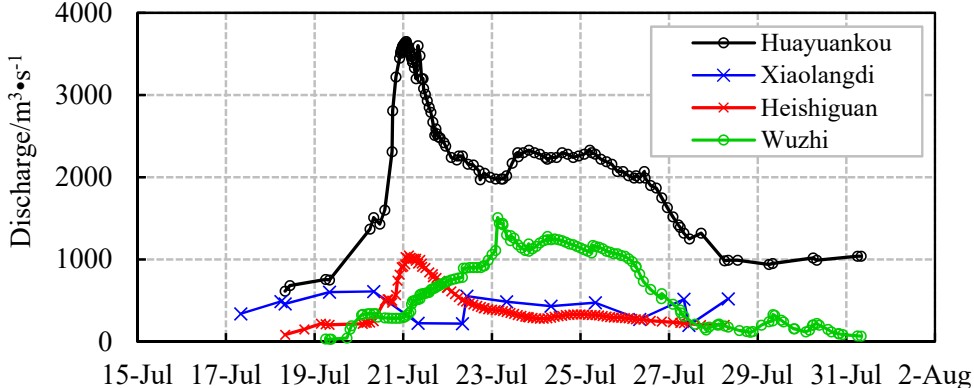

**Figure 2.** The runoff composition diagram of "7.21" flood at Huayuankou station.

In this paper, the time interval rainfall data of 980 rainfall stations (shown in Figure 3) during the "7.20" rainstorm period in 2021 were collected to draw the rainfall isogram. The long-series annual maximum 24 h and annual maximum 6 h rainfall data of ten reference rainfall stations were collected. The locations of 42 rainfall stations with the maximum 24 h rainfall greater than 300 mm and 46 rainfall stations with the maximum 6 h rainfall greater than 100 mm are shown in Figure 3.

*2.3. Method*

The ultimate aim of the hydrology frequency calculation is to determine the design value corresponding to the given design frequency. It stipulates that the line type of the frequency curve generally adopts Pearson-III distribution in the code for calculation of design flood of water conservancy and hydropower project [48].

The P-III curve is an unsymmetrical single-peak and positive partial curve with one infinite end. The probability density function of the Pearson-III curve is written as follows [49,50]:

$$f(x) = \frac{\beta^\alpha}{\Gamma(\alpha)}(x - a_0)^{\alpha-1}e^{-\beta(x-a_0)}dx \tag{1}$$

where: $\Gamma(\alpha)$ is the Gamma function, and $\alpha$, $\beta$, and $a_0$ are the shape, scale, and position parameters of the P-III distribution ($\alpha > 0$, $\beta > 0$).

After the integration, standardized transformation, and simplification for Formula (1), the equation for calculating the design value of the given design frequency can be deduced as:

$$
\begin{aligned}
X_P &= \overline{x} \times (1 + C_V \Phi_P) \\
&= \overline{x}\left(1 + C_V\left(\frac{C_S \times T_P}{2} - \frac{2}{C_S}\right)\right) \\
&= \overline{x}\left(1 + \frac{C_V \times C_V \times C_S/C_V \times T_P}{2} - \frac{2C_V}{C_V \times C_S/C_V}\right) \\
&= \overline{x}\left(1 + \frac{C_V{}^2 \times C_S/C_V \times T_P}{2} - \frac{2}{C_S/C_V}\right) \\
&= \overline{x}\left(1 + \frac{C_V{}^2 \times C_S/C_V \times \text{Gammainv}(1-P,\alpha,\beta)}{2} - \frac{2}{C_S/C_V}\right) \\
&= \overline{x}\left(1 + \frac{C_V{}^2 \times C_S/C_V \times \text{Gammainv}\left(1-P,4/C_S{}^2,\beta\right)}{2} - \frac{2}{C_S/C_V}\right)
\end{aligned}
\tag{2}
$$

where:

$$
\Phi_P = C_s/2 \times T_P - 2/C_S \tag{3}
$$

$$
T_P = \text{Gammainv}(1 - P, \alpha, \beta) \tag{4}
$$

$$
\alpha = 4/C_S^2, \ \beta = 1 \tag{5}
$$

where $P$ is the given design frequency, $X_P$ is the design value of the given design frequency, and Gammainv is the inverse function of the Gamma function. The macro command programming in Excel was used to find out the mean value, $\overline{x}$, variation coefficient, $C_V$, coefficient of skew, $C_S$, and Gammainv ($1 - p$, alpha and beta) of series $x_i$ ($i = 1, 2, \ldots, n$), and to draw the Pearson-III frequency curves of the series ($X_P$). The parameters of mean, $C_V$, and $C_S/C_V$ were determined by the estimation of the line-fitting method. According to Formula (2), the frequency corresponding to the rainfall or peak discharge was calculated.

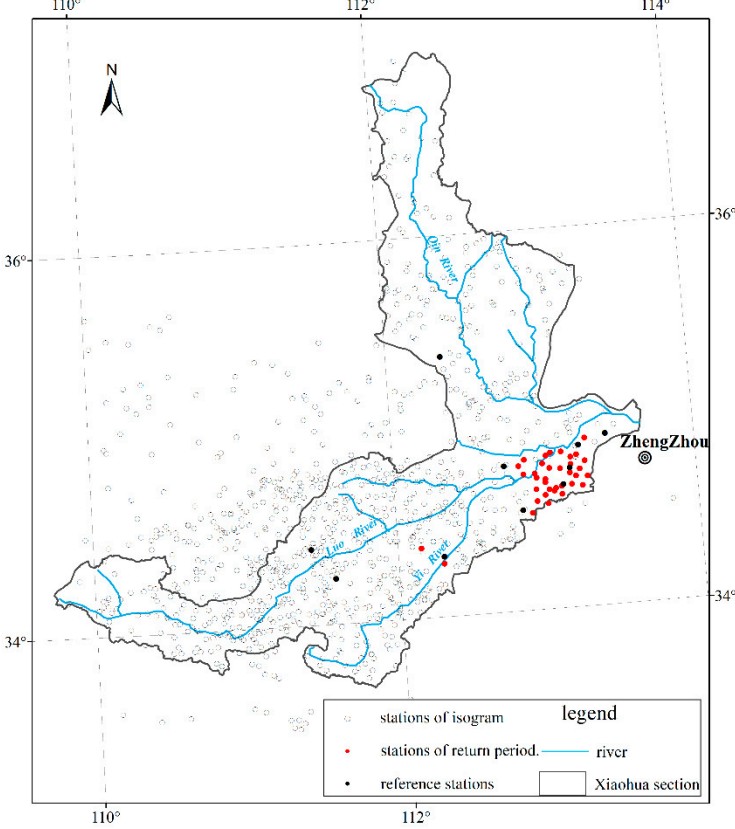

**Figure 3.** Location of rainfall stations.

## 3. Results and Discussion

### 3.1. "7.20" Rainstorm and Flood

The continuous heavy rain process occurred in the section from Xiaolangdi to Huayuankou on 18 to 22 July. There were 70 rainfall stations with cumulative rainfall over 400 mm. The number of stations with daily rainfall exceeding 50 mm was up to 762, and the maximum daily rainfall was 454.5 mm in Fenggou station. The top 20 daily rainfall events occurred on 20 July and 19 July, as shown in Table 1.

Affected by rainstorm, the peak flow of Heishiguan station in Yiluo River was 1050 m$^3$/s at 3:00 on 21 July, and that of Wuzhi station in Qin River was 1510 m$^3$/s at 3:12 on 23 July, which was the largest peak flow since 1982. After confluence of floods in Yiluo River, Qin River, and the Xiaohua main stream, the peak flow of Huayuankou station of the Yellow River reached 3650 m$^3$/s at 1:24 on 21 July (shown in Figure 3).

### 3.2. The Comparison of Historical and Current Maximum 24 h Rainfall

The maximum 24 h rainfall of "7.20" rainstorms of all ten reference stations was greater than that of the history series data (shown in Figure 4), and some stations were 2–3.6 times the historical maximum. The maximum 6 h rainfall of six reference stations was greater than the historical series data (shown in Figure 5).

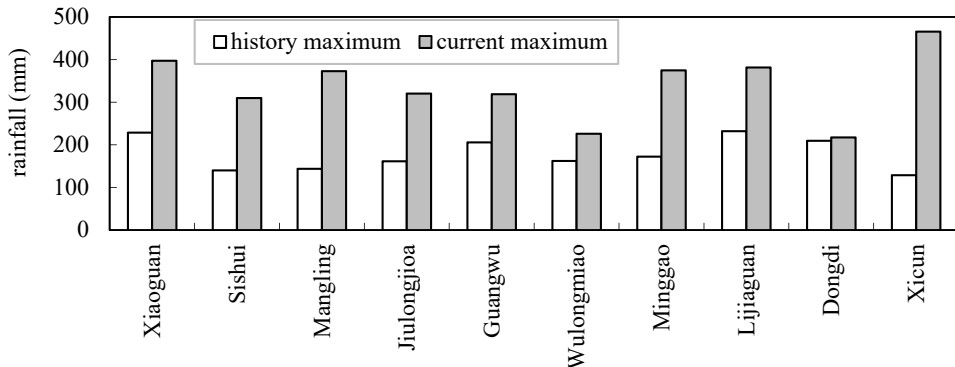

**Figure 4.** The maximum 24 h rainfall of ten reference stations between historical and "7.20" rainstorms.

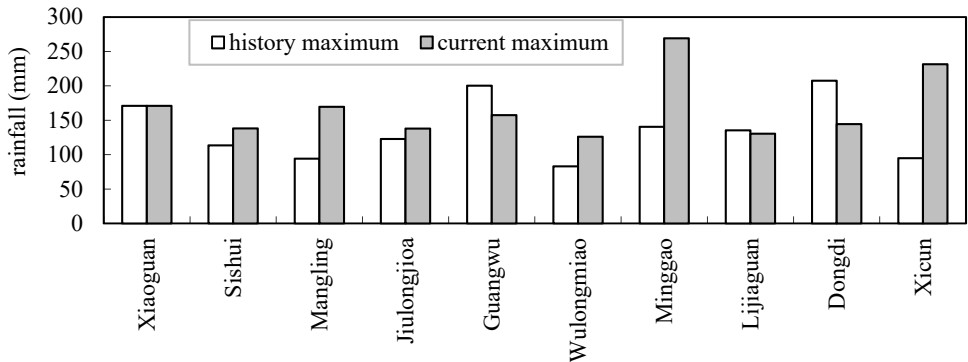

**Figure 5.** The maximum 6 h rainfall of ten reference stations between historical and "7.20" rainstorms.

### 3.3. The Analysis of Rainfall Isogram

The rainstorm covered the main stream of Xiaohua section, the lower region of the intersection of Yihe River and Luohe River, and the middle and lower reaches of Qinhe River.

The heavy rainstorm center was Liuhe station in the main Yellow River with the rainfall of 860 mm. The average rainfall in Xiaohua section was 180.9 mm on 18 to 22 July. It reached 395.1 mm in the main stream of the Xiaohua section of the Yellow River, while it was 315.7 mm in the region from Longmenzhen, Baimasi, to Heishiguan. It was 266.4 mm in the region from Runcheng to Wulongkou of Qinhe River, and 266.1 mm above Shanluping hydrological station in Danhe basin (shown in Figure 6 and Table 2).

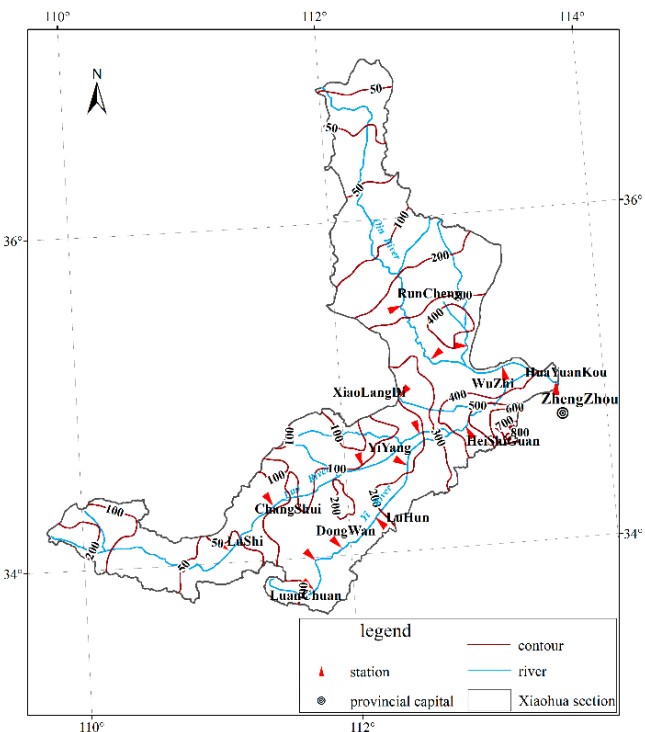

**Figure 6.** The isogram of rainfall and the location of reference stations in "7.20" rainstorms in the Xiaohua section.

### 3.4. The Frequency Curve and Return Period

The rainfall in two representative periods of the ten reference stations was analyzed by the P-III; frequency curve. The frequency curve was established in two cases: not considering the current rainfall, i.e., selecting the series up to 2020, and considering the current rainfall, i.e., the data in 2021 participating in the frequency analysis.

The Moment Method was used to estimate the parameters, and the deviation square sum minimum criterion was used to automatically optimize the line fitting. Finally, the parameters of mean, $C_V$, and $C_S/C_V$ were determined by the estimation of the line-fitting method. According to Formula (2), the return period of "7.20" rainfall for the 10 reference stations was calculated.

Taking Wulongmiao station as an example, the frequency curve of annual maximum 24 h rainfall is shown in Figures 7 and 8. The frequency analysis parameters of annual maximum 24 h rainfall of the reference stations are shown in Table 3.

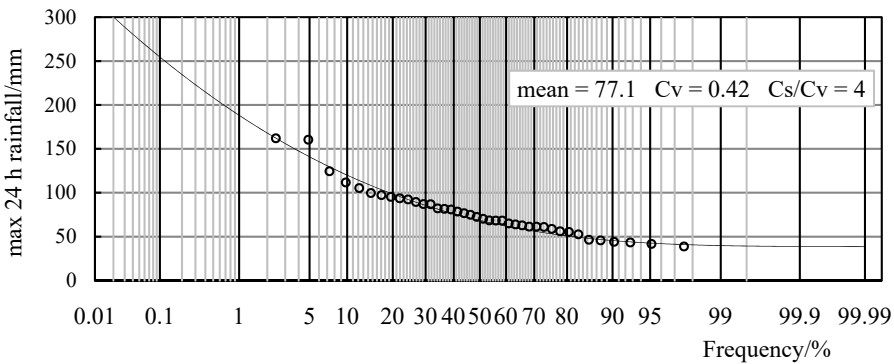

**Figure 7.** The frequency curve of the maximum 24 h rainfall of discontinuous 40 years in 1978~2020 at Wulongmiao station.

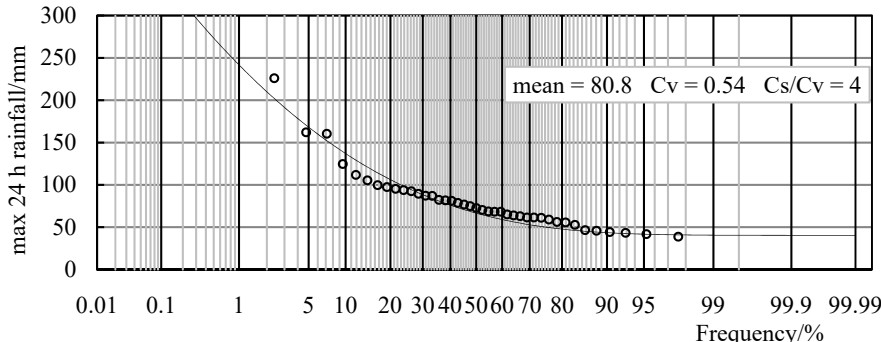

**Figure 8.** The frequency curve of the maximum 24 h rainfall of discontinuous 41 years in 1978~2021 at Wulongmiao station.

**Table 3.** Frequency analysis parameters and return period of maximum 24 h rainfall of "7.20" flood in 2021 for reference stations.

| No. | Station | Series/a | Parameters | | | Max/mm | Frequency/% | Return Period/a |
| --- | --- | --- | --- | --- | --- | --- | --- | --- |
| | | | Mean/mm | $C_v$ | $C_s/C_v$ | | | |
| 1 | Xiaoguan | 34 | 96.2 | 0.51 | 3.5 | 228.5 | 0.057 | 1754 |
| | | 35 | 104.8 | 0.65 | 3.5 | 397.4 | 0.602 | 166 |
| 2 | Sishui | 38 | 84.0 | 0.41 | 3 | 139.8 | 0.009 | 11,123 |
| | | 39 | 89.8 | 0.52 | 3 | 309.8 | 0.2056 | 486 |
| 3 | Mangling | 32 | 67.8 | 0.42 | 5 | 143.7 | 0.00113 | 88,730 |
| | | 33 | 77.1 | 0.85 | 5 | 372.8 | 0.951 | 105 |
| 4 | Jiulongjioa | 40 | 72.4 | 0.45 | 4 | 161.4 | 0.01174 | 8515 |
| | | 41 | 78.4 | 0.62 | 4 | 320.4 | 0.03733 | 268 |
| 5 | Guangwu | 30 | 76.5 | 0.53 | 4 | 205.5 | 0.1097 | 911 |
| | | 31 | 84.3 | 0.78 | 4 | 318.6 | 1.4908 | 67 |
| 6 | Wulongmiao | 40 | 77.1 | 0.42 | 4 | 162 | 0.2738 | 365 |
| | | 41 | 80.8 | 0.54 | 4 | 225.8 | 1.409 | 71 |
| 7 | Minggao | 47 | 77.3 | 0.47 | 4 | 172.4 | 0.0077 | 13,004 |
| | | 48 | 83.5 | 0.66 | 4 | 374.8 | 0.320 | 313 |
| 8 | Lijiaguan | 40 | 98.6 | 0.55 | 3 | 232 | 0.1285 | 778 |
| | | 41 | 105.5 | 0.64 | 3 | 381.6 | 0.582 | 172 |
| 9 | Dongdi | 38 | 84.9 | 0.56 | 3.5 | 209.3 | 2.246 | 45 |
| | | 39 | 88.3 | 0.59 | 3.5 | 217.4 | 3.127 | 32 |
| 10 | Xicun | 38 | 70.2 | 0.42 | 4 | 128.5 | - | - |
| | | 39 | 80.3 | 0.85 | 4 | 465.6 | 0.37 | 270 |

The frequency curves of annual maximum 6 h rainfall of the 10 reference stations were established. Taking Jiulongjiao and Minggao station as examples, the frequency curves are shown in Figures 9 and 10. The frequency analysis parameters of annual maximum 6 h rainfall of the reference stations are shown in Table 4.

According to the established frequency curve parameters, the rainfall return period of reference stations was calculated. It can be obtained that current rainfall, that is "7.20" rainfall, has a very obvious impact on the rainfall return period.

The maximum 24 h rainfall of "7.20" rainstorms for the 10 reference stations was the maximum of the whole series. When establishing the frequency curve, if this value is considered, the maximum return period is 486 years for Sishui station. Otherwise, the return periods of Sishui, Mangling, Minggao, and Xicun stations exceed 10,000 years. The return periods of eight reference stations showed significant differences of orders of magnitude between considering and removing the "7.20" rainfall (shown in Table 3).

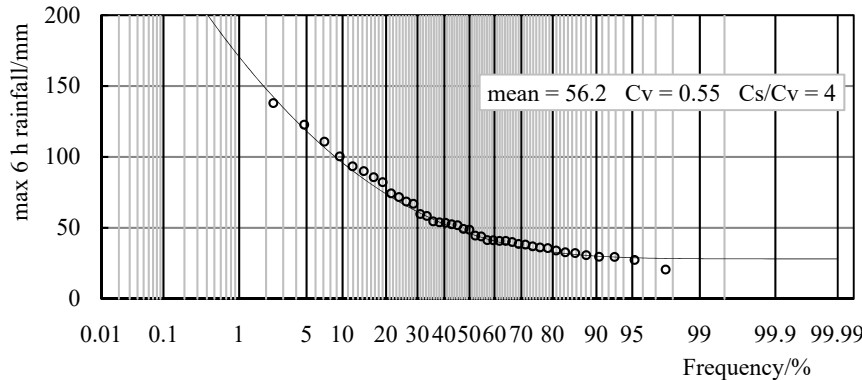

**Figure 9.** The frequency curve of the maximum 6 h rainfall of discontinuous 41 years in 1978~2021 at Jiulongjiao station.

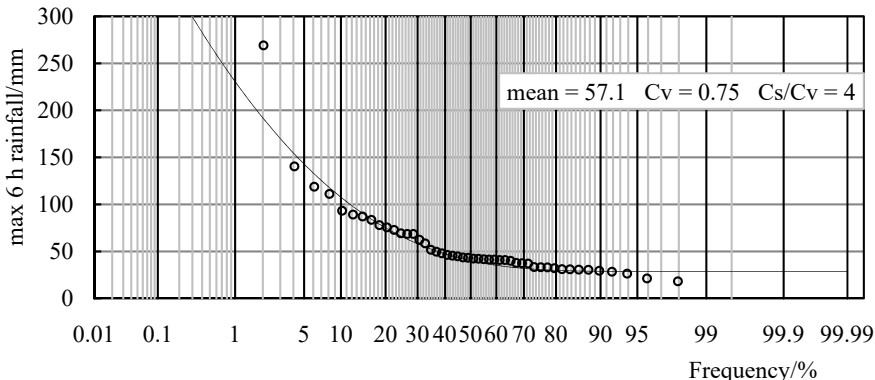

**Figure 10.** The frequency curve of the maximum 6 h rainfall of discontinuous 48 years in 1964~2021 at Minggao station.

**Table 4.** Frequency analysis parameters and return period of maximum 6 h rainfall of "7.20" flood in 2021 for reference stations.

| No. | Station | Series/a | Parameters | | | max/mm | Frequency/% | Return Period/a |
| | | | Mean/mm | $C_v$ | $C_s/C_v$ | | | |
|---|---|---|---|---|---|---|---|---|
| 1 | Xiaoguan | 34 | 64.2 | 0.52 | 4 | 171 | 6.17 | 16 |
| | | 35 | 65.9 | 0.55 | 4 | 171 | 7.41 | 13 |
| 2 | Sishui | 38 | 63.5 | 0.45 | 3 | 113.6 | 2.19 | 46 |
| | | 39 | 65.5 | 0.44 | 3 | 138.2 | 2.41 | 41 |
| 3 | Mangling | 32 | 47.1 | 0.45 | 5 | 94.2 | 0.151 | 662 |
| | | 33 | 50.8 | 0.56 | 5 | 169.6 | 0.85 | 118 |
| 4 | Jiulongjioa | 40 | 54.1 | 0.51 | 4 | 122.8 | 1.79 | 56 |
| | | 41 | 56.2 | 0.55 | 4 | 138 | 2.71 | 37 |
| 5 | Guangwu | 30 | 60.6 | 0.67 | 4 | 200.2 | 3.63 | 28 |
| | | 31 | 63.7 | 0.78 | 4 | 157.4 | 5.47 | 18 |
| 6 | Wulongmioa | 40 | 49.8 | 0.32 | 4 | 83.1 | 0.122 | 820 |
| | | 41 | 51.7 | 0.42 | 4 | 126 | 1.01 | 99 |
| 7 | Minggao | 47 | 52.6 | 0.56 | 4 | 140.5 | 0.0353 | 2937 |
| | | 48 | 57.1 | 0.75 | 4 | 269.2 | 0.506 | 198 |
| 8 | Lijiaguan | 40 | 65.2 | 0.49 | 3 | 135.4 | 4.49 | 22 |
| | | 41 | 66.8 | 0.5 | 3 | 130.4 | 5.27 | 19 |
| 9 | Dongdi | 38 | 57.1 | 0.66 | 3.5 | 207.5 | 3.77 | 27 |
| | | 39 | 59.4 | 0.76 | 3.5 | 144.4 | 5.59 | 18 |
| 10 | Xicun | 38 | 52.1 | 0.45 | 4 | 94.8 | 0.0113 | 8850 |
| | | 39 | 56.7 | 0.62 | 4 | 231.4 | 0.375 | 267 |

Among the 10 reference stations, Minggao station had the largest proportion between the maximum 6 h rainfall of "7.20" rainstorms and the maximum 6 h value of historical series. Taking Minggao station as an example, when establishing the frequency curve, if this value is not considered, the return period is about 3000 years, and if this value is considered, the return period is about 200 years (shown in Table 4).

According to the frequency curve parameters of reference stations, the rainfall return period of nearby rainfall stations can be calculated (shown in Table 5).

**Table 5.** The return period of rainfall in different periods of each station in "7.20" rainstorm.

| No. | River | Station | Max 24 h (>300 mm) | | Max 6 h (>100 mm) | |
| --- | --- | --- | --- | --- | --- | --- |
| | | | Rainfall/mm | Return Period/*a* | Rainfall/mm | Return Period/*a* |
| 1 | Yiluo River | Huancuiyu | 643.5 | 4424 | 257.5 | 440 |
| 2 | Yiluo River | Liuhe | 572.5 | 1725 | 247.5 | 340 |
| 3 | Yiluo River | Duanhecun | 539 | 1105 | 232.5 | 231 |
| 4 | Yiluo River | Xinzhong | 536 | 1062 | 216.5 | 153 |
| 5 | Yiluo River | Fenggou | 533 | 1020 | 246 | 327 |
| 6 | Yiluo River | Hegou | 476.5 | 480 | 233.5 | 237 |
| 7 | Yiluo River | Zhulin | 476 | 4051 | 181 | 137 |
| 8 | Yiluo River | Xicun | 465.6 | 270 | 231.4 | 267 |
| 9 | Yiluo River | Heluozhen | 440 | 295 | 234.5 | 243 |
| 10 | Yiluo River | Gongchuan | 433 | 1922 | 173.5 | 109 |
| 11 | Yiluo River | Tiejianglu | 428.5 | 253 | 166.5 | 42 |
| 12 | Yiluo River | Zhangyao | 417.5 | 157 | 195 | 239 |
| 13 | Yiluo River | Shanhua | 413 | 151 | 196.5 | 249 |
| 14 | Yiluo River | Xiayu | 412.5 | 204 | 134.5 | 18 |
| 15 | Yiluo River | Liangshuiquan | 408.5 | 194 | 162 | 37 |
| 16 | Yiluo River | Changzhuang | 408.5 | 194 | 168.5 | 44 |
| 17 | Yellow River | Xiaoguan | 397.4 | 166 | 123.8 | 13 |
| 18 | Yiluo River | Shennan | 396 | 164 | 227 | 201 |
| 19 | Yiluo River | Honghe | 394.5 | 160 | 132 | 17 |
| 20 | Yiluo River | Nanhedu | 383.5 | 138 | 215.5 | 149 |
| 21 | Yiluo River | Lijiaguan | 381.6 | 172 | 130.4 | 19 |
| 22 | Yiluo River | Shidonggou | 377.5 | 128 | 173.5 | 50 |
| 23 | Yiluo River | Minggao | 374.8 | 313 | 269.2 | 198 |
| 24 | Yiluo River | Heishiguan | 374.4 | 107 | 172.6 | 128 |
| 25 | Yellow River | Mangling | 372.8 | 105 | 169.6 | 118 |
| 26 | Yiluo River | Jiuhouxiang | 362.5 | 260 | 257 | 160 |
| 27 | Yiluo River | Shanchuan | 360 | 101 | 149 | 27 |
| 28 | Yiluo River | Zhaogou | 357.5 | 92 | 171.5 | 125 |
| 29 | Yiluo River | Hetaoyuan | 353.5 | 92 | 119 | 12 |
| 30 | Yellow River | Gaoshan | 353.5 | 92 | 154.5 | 31 |
| 31 | Yiluo River | Zhanjie | 352.5 | 91 | 181 | 61 |
| 32 | Yiluo River | Didong | 335.5 | 350 | 159 | 70 |
| 33 | Yiluo River | Wuluo | 333.5 | 71 | 159 | 35 |
| 34 | Yiluo River | Mihezhen | 333.5 | 71 | 134.5 | 18 |
| 35 | Yiluo River | Wanggou | 332 | 73 | 145 | 59 |
| 36 | Yiluo River | Guandimiao | 331 | 323 | 146.5 | 48 |
| 37 | Yiluo River | Jiulongjiao | 320.4 | 268 | 138 | 37 |
| 38 | Yellow River | Guangwu | 318.6 | 67 | 157.4 | 18 |
| 39 | Yiluo River | Fangluo | 316 | 66 | 127.5 | 12 |
| 40 | Yiluo River | Jiajinkou | 315.5 | 55 | 140.5 | 22 |
| 41 | Yiluo River | Shecun | 315 | 55 | 112.5 | 11 |
| 42 | Yiluo River | Sishui | 309.8 | 486 | 138.2 | 41 |
| 43 | Yiluo River | Lifeng | - | - | 199 | 57 |
| 44 | Yiluo River | Baiyu | - | - | 186.5 | 162 |
| 45 | Qin River | Dongdi | - | - | 144.4 | 18 |
| 46 | Yiluo River | Wulongmiao | - | - | 126 | 99 |

One is the return period of 42 rainfall stations with the maximum 24 h rainfall greater than 300 mm. The maximum 24 h rainfall of Huancuiyu station is 643.5 mm, with a return period of about 4000 years, ranking first among all the stations. The return period of rainfall between 400 and 600 mm is about 150~4000 years, and the return period of rainfall between 250 and 400 mm is about 40~150 years.

The other is the return period of 46 rainfall stations with the maximum 6 h rainfall greater than 100 mm. The maximum 6 h rainfall of Minggao station is 269.2 mm, with a return period of about 200 years, ranking first among all the stations. The return period of 257.5 mm of rainfall of Huancuiyu station is about 400 years, and the return period of rainfall of other stations between 100 and 250 mm is about 10~300 years.

## 4. Conclusions

The impact of current maximum value, the "7.20" rainstorm in this study, on the return period of rainfall is very obvious. The maximum 24 h rainfall of "7.20" rainstorms for the ten reference stations were the maximum values of the respective series. When establishing the frequency curve, if this value was considered, the maximum return period was 486 years for Sishui station. If this value was not considered, the return period of Sishui, Mangling, Minggao, and Xicun stations exceeded 10,000 years.

Among the ten reference stations, Minggao station had the largest proportion between the maximum 6 h rainfall of "7.20" rainstorms and the maximum value of historical series. Taking Minggao station as an example, the return period was about 200 years when considering the value to establish the frequency curve, otherwise it was about 3000 years.

When the current rainfall is the maximum value of data series of the analyzed rainfall stations, we suggest that the current maximum value should be considered when establishing the frequency curve and analyzing the return period of rainfall.

**Author Contributions:** Conceptualization, S.J.; Data curation, S.J., S.G. and W.H.; Formal analysis, S.J.; Investigation, S.G. and W.H.; Methodology, S.J.; Project administration, S.G.; Software, S.G.; Supervision, S.J.; Validation, S.G. and W.H.; Visualization, S.G.; Writing—original draft, S.J.; Writing—review & editing, S.G. and W.H. All authors have read and agreed to the published version of the manuscript.

**Funding:** This research was funded by the National Key Research and Development Program of China grant number 2021YFC3201101.

**Institutional Review Board Statement:** Not applicable.

**Informed Consent Statement:** Not applicable.

**Data Availability Statement:** Not applicable.

**Conflicts of Interest:** The authors declare no conflict of interest.

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
