# Peer review of "Analysis on the Return Period of “7.20” Rainstorm in the Xiaohua Section of the Yellow River in 2021"

_water, doi:10.3390/w14152444_

Round 1
Reviewer 1 Report
Comments-questions (to be clarified in the text)
1. Abstract and Introduction: Under "P-III distribution" is the "Pearson III distribution" meant?
2. Equations (2)-(5): (a) It should be explicitly explained what p and XP are. (b) Is there a difference between p and P? (c) What is under "Gammainv" meant? (d) The parameters "mean", "standard deviation", Cv and Cs are usually determined from the available data. (e) The computational steps should be described in detail.
3. Figures 4 and 5: The signs for historical and current maximum are not clear.
4. References: They should be written in a uniform way, for example: (a) no capitals for the surnames of the authors should be used, except for the first letter; (b) only the initials of the first names of the authors should be written.
5. See annotated manuscript!

Author Response
1. Abstract and Introduction: Under "P-III distribution" is the "Pearson III distribution" meant?
Yes, the "P-III distribution" is the "Pearson III distribution"
2. Equations (2)-(5): (a) It should be explicitly explained what p and XP are. (b) Is there a difference between p and P? (c) What is under "Gammainv" meant? (d) The parameters "mean", "standard deviation", Cv and Cs are usually determined from the available data. (e) The computational steps should be described in detail.
P is the given design frequency, XP the design value of the given design frequency, Gammainv the inverse function of the Gamma Function. p and P are the same meaning, which we have unified in the manuscript. The Pearson III distribution is a common distribution function so we simplified the calculation steps. See lines 139-140.
3. Figures 4 and 5: The signs for historical and current maximum are not clear.
The Figures 4 and 5 are revised.
4. References: They should be written in a uniform way, for example: (a) no capitals for the surnames of the authors should be used, except for the first letter; (b) only the initials of the first names of the authors should be written.
The References are revised.
5. See annotated manuscript!
Revised.
Reviewer 2 Report
From the point of view of the research and methodology used, this approach is synthetic, correctly made and with a series of pertinent conclusions. From my point of view, a brief analysis of the synoptic situation that led to the genesis of the quantities of precipitation that fell in the study area should be performed. This will strengthen research results and demonstrate the climate change pact.
Author Response
Thanks for your comment! However, the analysis of the synoptic situation is beyond the authors' ability. We mainly analyze the frequency of rainfall and flood in this paper.
Round 2
Reviewer 1 Report
1. Line 145: More details are needed for the fitting line method.
2. References: The initials of the first names only should be written.
3. See annotated manuscript! There are many points to be improved.

Author Response
- Line 145: More details are needed for the fitting line method.
The detailed calculation processes are added in line 136.
- References: The initials of the first names only should be written.
The References have been revised.
- See annotated manuscript! There are many points to be improved.
Other errors have been revised.

This manuscript is a resubmission of an earlier submission. The following is a list of the peer review reports and author responses from that submission.